# End-to-End Entanglement Generation Strategies: Capacity Bounds and Impact on Quantum Key Distribution

Antonio Manzalini [1,*] and Michele Amoretti [2]

1 Technology Communication and Standardization, TIM—Telecom Italia, 10148 Torino, Italy
2 Department of Engineering and Architecture, University of Parma, 43124 Parma, Italy; michele.amoretti@unipr.it
* Correspondence: antonio.manzalini@telecomitalia.it

**Abstract:** A first quantum revolution has already brought quantum technologies into our everyday life for decades: in fact, electronics and optics are based on the quantum mechanical principles. Today, a second quantum revolution is underway, leveraging the quantum principles of superposition, entanglement and measurement, which were not fully exploited yet. International innovation activities and standardization bodies have identified four main application areas for quantum technologies and services: quantum secure communications, quantum computing, quantum simulation, and quantum sensing and metrology. This paper focuses on quantum secure communications by addressing the evolution of Quantum Key Distribution (QKD) networks (under early exploitation today) towards the Quantum-ready networks and the Quantum Internet based also on entanglement distribution. Assuming that management and control of quantum nodes is a key challenge under definition, today, a main obstacle in exploiting long-range QKD and Quantum-ready networks concerns the inherent losses due to the optical transmission channels. Currently, it is assumed that a most promising way for overcoming this limitation, while avoiding the presence of costly trusted nodes, it is to distribute entangled states by means of Quantum Repeaters. In this respect, the paper provides an overview of current methods and systems for end-to-end entanglement generation, with some simulations and a discussion of capacity upper bounds and their impact of secret key rate in QKD systems.

**Keywords:** quantum communications; Quantum Key Distribution; entanglement; Quantum Repeaters

## 1. Introduction

Today, telecommunications are witnessing a pervasive diffusion of ultra-broadband fixed-mobile connectivity, the deployment of 5G networks and service platforms and a wide adoption of Artificial Intelligence (AI). This Digital Transformation is expected to bring far reaching techno-economic impacts on our society by providing any sort of digital services and applications.

At the same time, the sustainability of future networks and services scenarios will have to face several challenges, such as the transmission and processing of enormous and increasing quantity of data with ultra-low latency, automation of management and control processes, the fulfillment of strict requirements of resilience, security, and privacy, optimization of energy consumption, and so on. Moreover, the reliance of the ongoing electronic trend on packing more transistors into the same amount of silicon is reaching the very boundaries of what is physically possible, leading to some worries on the progress of this pace of innovation.

In this context, a rethinking of new ways for communications and computing has already started, and quantum technologies are likely to address some of these challenges and limitations. As a matter of fact, there is a new impressive growth of interest in quantum technologies, with several investments from public and private organizations worldwide targeting new horizons of applications.

A first quantum revolution already brought quantum technologies into our everyday life decades ago. Chips for computers and smart-phones, systems for medical imaging (Nuclear Magnetic Resonance, Positron Emission Tomography), LED and lasers, etc., are all based on technologies exploiting the quantum mechanics principles.

Today, a *second quantum revolution* is underway, leveraging on the three quantum principles of superposition, entanglement and measurement, which were not engineered during the previous phase. Quantum technologies are progressing quickly, and it is safe to predict that a second wave of quantum technologies could potentially have a major impact in many markets, ranging from Telecom and ICT, to Medicine, to Finance, to Transportation, and so on. International innovation activities and standardization bodies are aligned in identifying four main application areas of quantum technologies and services: quantum communications, quantum computing, quantum simulations, and quantum sensing and metrology.

The area of quantum communications includes two main sub-domains: quantum-safe communications, and long-distance "teleportation" of qubits. Quantum-safe communications leverage on systems such as Quantum Key Distribution (QKD) [1–3] and Quantum Random Number Generators (QRNG) [4], which today are becoming mature for concrete industrial applications.

Quantum computing [5–7] concerns the exploitation of the three principles of superposition, (hyper)entanglement and measurements, to speed up over classical computers in solving complex optimization and combinatorial problems.

Quantum simulations [8,9] concern all those applications where well-controlled quantum systems are used to simulate the behavior of other systems, which are less accessible and more complex for direct simulations.

Quantum sensing and metrology [10] includes those applications where high sensitivity of quantum systems to environmental influences can be exploited to measure physical properties and timing with more precision (e.g., magnetic and heat sensors, gravimeters, GPS-free navigators, clocks).

Importantly, it is likely that in the medium-long term, this second wave of quantum technologies will bring network evolution towards the *Quantum Internet* [11–14], which is a global network exploiting the principles of Quantum Physics for transmitting, processing, and storing qubits, the units of quantum information. Communications capabilities and services of the Quantum Internet basically leverage on the distribution of entangled quantum states, quantum channels, and among remote quantum nodes and devices. In particular, quantum channels work in synergy with classical links.

Assumed that management and control of quantum nodes is a key challenge under definition, today, a main obstacle in exploiting long-range QKD and Quantum-ready networks concerns the inherent losses due to the optical transmission channels. Currently, it is assumed that the most promising way for overcoming this obstacle is based upon the adoption of Quantum Repeaters (QRs). This approach is useful also from another perspective: in quantum-safe communications, the adoption of untrusted QRs is an appealing alternative to the use of intermediate trusted nodes, paving the way for more efficient, less expensive QKD networks.

As it is not clear yet which QR technology will succeed, a robust strategy for designing and analyzing QR-based quantum networks is using discrete event simulation. In this sense, the main scope of this paper is discussing the capacity upper bounds of QR networks with respect to entanglement generation and presenting a study of a simple network model with a certain number of QRs using the NetSquid simulation.

The main contributions provided by the paper can be summarized in terms of:

1. Offering a brief introduction on the evolution towards the Quantum Internet. Specifically, the three main research and innovation avenues are mentioned: quantum transmission, networking protocols and management-control paradigms;

2. Providing an overview of current strategies for end-to-end entanglement generation, with the discussion of capacity upper bounds and their impact of secret key rate in QKD systems;
3. Presenting the simulation of a simple quantum network model with a certain number of QRs.

**2. Toward the Quantum Internet**

The main principles of the Quantum Internet have been recently summarized by an IRTF Draft [13].

It should be noted that while, in the classical internet, bits can be duplicated within a node or among the different nodes of a network, this is not valid for the Quantum Internet because of the no-cloning theorem [5], which forbids any possibility of duplicating an unknown qubit. This means that although a qubit can be transmitted directly to a remote node via a fiber link, attenuation or noise can degenerate the qubit and above a certain degree of degradation the quantum information cannot be recovered via a measuring process, amplification, or a duplication.

In fiber-based telecommunication networks, the most promising way to overcome the distance limits is to adopt QRs to generate end-to-end entanglement to be used for quantum state teleportation. Recently, significant progress has been made both theoretically and experimentally, however, still some technological obstacles remain in building practical QRs, with proper levels of performance.

This is not the only area where innovation progresses are required to pave the way towards the Quantum Internet. In fact, it should also be mentioned that this evolution will require a re-design of the network protocols dictated by the use of quantum technologies, at the data link and network layer, and their integration. As a matter of fact, a full integration of classical and quantum network and service resources is crucial for the Quantum Internet [14], also from the management and control viewpoints.

*2.1. Network Protocols*

Physical layer issues, e.g., from heat, electromagnetic effects or quantum-mechanical processes, can occasionally flip or randomize the state of a qubit, potentially derailing quantum computations. Moreover, optical transmission losses can jeopardize entanglement distribution. To cope with these problems, quantum error correction and QRs are required [14].

Regarding the data link layer issues, the no-broadcasting theorem (corollary of the no-cloning theorem) prevents quantum information from being transmitted to more than a single destination. This is a fundamental difference with respect to current networks, where broadcasting is exploited in layer-2 and -3 functionalities. In this sense, the link layer of the Quantum Internet then must be re-designed to allow multiple quantum devices to be connected to a single quantum channel (e.g., a fiber link) [14].

Concerning the network layer, entanglement distribution determines the connectivity to perform quantum state teleportation between quantum devices. Hence, novel quantum routing metrics and protocols are needed to ensure effective entanglement-aware path selection. Since teleportation destroys entanglement due to Bell-State Measurement (BSM), if another qubit needs to be teleported, a new entangled pair must be created and distributed between the source and the destination [14].

*2.2. Management and Control*

Software Defined Networks (SDN) technologies offer very flexible ways to manage and control functions, resources and services of a telecommunications network. In general, SDN allows the management and optimization of the entire infrastructure from a logically centralized element, usually denoted as SDN controller. Moreover, programmability and flexibility brought by SDN technologies drastically reduce the time and effort of integrating new devices and technologies into the network.

It should also be considered that one major obstacle hindering the exploitation of quantum computing and communications technologies is that the industry has not yet consolidated around one type of quantum hardware technology.

In this context, the definition of a Quantum-Hardware Abstraction Layer (Q-HAL) would decouple the hardware from the software and service developments, thus allowing Applications and Services Developers to start using the abstraction functionalities provided by the underneath quantum hardware even if under consolidation.

A Q-HAL would simplify and speed-up the creation and use of quantum platforms, services, and applications necessary to develop industrial quantum ecosystems, either for communications or computing.

Practically, as an example, a Q-HAL would provide unified northbound quantum Application Programming Interfaces (APIs) for the higher layers, decoupling from the different types of quantum hardware technologies (e.g., trapped ions, superconducting qubits, silicon photonics qubits) for quantum communications and computing services.

This is important also for another reason: any quantum node or system of a Quantum Communication Infrastructure (QCI) needs to be properly configured, managed, and monitored to effectively operate and support interworking with classic nodes/systems. In this sense, there is a strict need for defining data models representing various aspects of the networked quantum sub-systems and components (or devices). This will bring interoperability and plug-and-play capabilities. As an example, an option could be using the YANG data model, which is standard language widely a used in the telecommunication domain for describing and managing devices on a network.

In general, the activities of definition, modeling, and standardization of a Quantum-HAL—for a QCI—will require coordinated and joint efforts including, where appropriate, existing projects, industry bodies and standard fora (e.g., CEN-CENELEC, GSMA, ETSI, IEEE, etc.) active in the area.

## 3. End-to-End Entanglement Generation

In this section, we discuss the importance of entanglement as a resource for quantum communication, and we present QR networks with special attention to performance indicators.

### 3.1. Entanglement as a Resource

Distributed entanglement is a precious quantum resource, as it finds applicability in many protocols. In the following, we shortly recap some of the most valuable entangled states, together with their main applications.

Bell states, also known as Einstein–Podolsky–Rosen pairs [5], are maximally entangled two-qubit states defined as

$$|\beta_{00}\rangle = |\phi^+\rangle = \frac{|00\rangle + |11\rangle}{\sqrt{2}}, \tag{1}$$

$$|\beta_{01}\rangle = |\psi^+\rangle = \frac{|01\rangle + |10\rangle}{\sqrt{2}}, \tag{2}$$

$$|\beta_{10}\rangle = |\phi^-\rangle = \frac{|00\rangle - |11\rangle}{\sqrt{2}}, \tag{3}$$

$$|\beta_{11}\rangle = |\psi^-\rangle = \frac{|01\rangle - |10\rangle}{\sqrt{2}}. \tag{4}$$

Bell states are an indispensable resource for key protocols like quantum state teleportation, QKD, cluster state preparation, and superdense coding. Moreover, Bell pairs play a significant role in distributed quantum computing, by supporting the implementation of remote operations (telegates) involving data qubits of two different devices [15,16].

Greenberger–Horne–Zeilinger (GHZ) states, generalizing Bell states to more than two qubits, have several practical applications, including quantum machine learning [17,18]. Finally, cluster states are a fundamental resource for measurement-based quantum compu-

tation [19–21]. In this model, a computation is described by a set of measurement angles on an entangled state, following what is called a measurement pattern. The state preparation phase is technologically more challenging than the computation one.

The latency associated with entanglement generation over long distances does not affect the latency of quantum state distribution implemented by means of teleportation, provided that entangled states can be buffered. To this purpose, good quantum memories are necessary. A shared Bell pair can be distributed between two nodes and held in their quantum memories until needed for the execution of the teleportation protocol. The latency of transmitting a quantum state between the two nodes is determined entirely by the latency of the classical channel, which is used to communicate the local corrections required to complete the teleportation protocol [5].

### 3.2. Quantum Repeater Networks

The distribution of quantum states over long distances is limited by photon loss. To solve this problem, QR protocols can be used to create long-distance entanglement from shorter-distance entanglement via entanglement swapping [12,22]. The field of QR networks is growing fast, with different competing repeater designs of ever-increasing sophistication.

In QR networks, there are three main operations: (i) the creation of entangled links between adjacent repeater nodes (entanglement distribution); (ii) the amelioration of the quality of entanglement between nodes (entanglement purification); and (iii) the joining of adjacent entangled links (entanglement swapping), by means of hierarchical or simultaneous BSMs. In Figure 1, the case of hierarchical BSMs is illustrated.

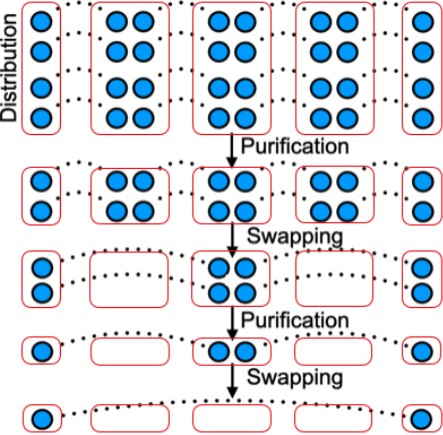

**Figure 1.** Operations of a QR network with hierarchical BSMs: in the initial entanglement distribution, multiple Bell states are established across each pair of adjacent repeater nodes; then, purification and swapping iterations are executed, until the end-to-end entangled state has been generated.

In entanglement distribution between adjacent repeater nodes, channel losses are the dominant error source. Let us consider two repeater nodes characterized by distance $L$, attenuation length of the channel $L_0$ ($\simeq 22.5$ km in the optical fiber), and detector efficiency $p_{det}$. Ignoring the source and coupling efficiencies, we can express the heralded probability of success [12] as

$$p_{ED} = \frac{e^{-L/L_0} p_{det}^2}{2}.$$ 

(5)

Therefore, the time to generate a distributed entangled pair between two adjacent repeater nodes is

$$T_{\text{adj}} \simeq \frac{L}{c p_{ED}}$$

(6)

where $c$ is the speed of light in the channel (close to $2 \times 10^8$ m/s in the optical fiber).

If the state $|\psi\rangle$ of a quantum system is known exactly, we say the system is in a *pure state*. A quantum system may be in mixed state, i.e., a statistical ensemble of several quantum states $|\psi_i\rangle$ (not necessarily orthogonal), with respective probabilities $p_i$. A mixed state is fully described by the density operator defined by the following equation:

$$\rho = \sum_i p_i |\psi_i\rangle\langle\psi_i| \tag{7}$$

A popular measure of the closeness of two density operators is the *fidelity*. The most widely adopted definition of fidelity is Uhlmann–Jozsa's one:

$$F(\rho, \sigma) = \left(\mathrm{tr}\sqrt{\sqrt{\rho}\sigma\sqrt{\rho}}\right)^2. \tag{8}$$

In the context of quantum networks, fidelity is commonly used to evaluate the quality of generated/transmitted quantum states with respect to their pure state representations.

Entanglement purification is used to provide higher-fidelity entangled states, starting from those produced in the distribution phase. To obtain a single high-quality Bell state, multiple low-quality Bell states are consumed. Perfection is asymptotically approached by repeating the protocol.

Entanglement purification protocols are a sort of quantum error detection codes: both parties perform local operations and end up with a measurement; depending on the outcome, the resulting Bell state has a higher fidelity or is discarded. Therefore, the protocols are probabilistic.

Finally, entanglement swapping is the process of taking two Bell pairs $\rho_{a_1a_2}$ and $\rho_{b_1b_2}$ with fidelity $F$, performing a Bell state measurement between the qubits $a_2$ and $b_2$, thus obtaining the state $\rho_{a_1b_1}$ with fidelity $F' < F$, where $F' = F^2 + (1-F)^2 \simeq F^2$. With $N$ links, the final fidelity would scale as $F^N$. For high-fidelity end-to-end, several purification and swapping iterations have to be executed.

### 3.3. Capacity of a QR Chain

The number of end-to-end Bell states (ebits) generated per second is denoted as *capacity*.

For a pure-loss optical channel, with direct photon transmission, the exact expression of the entanglement generation capacity is

$$R_{\text{direct}} = -\log_2(1-\eta) \quad [\text{ebit/s}], \tag{9}$$

called the PLOB bound [23], where $\eta = e^{-\alpha L}$ is the transmissivity of the optical fiber, scaling exponentially with range $L$.

For a QR chain, the capacity can be estimated as

$$R = \frac{1}{TM} \quad [\text{ebit/s}] \tag{10}$$

where

- $T(n, k, L_{tot})$ is the time to generate a Bell state over the total distance $L_{tot}$, using an $n$-nested QR configuration with $k$ rounds of purification for each nesting level; this means that the number of intermediate QR nodes is $2^{n-1}$, and the distance between nodes is $L = L_{tot}/2^n$;
- $M(k, n)$ is the number of quantum memories used; discounting $R$ with $M$ provides a fairer comparison when different purification protocols are used [12].

Most of the repeater operations are probabilistic in nature (though they may be heralded), and classical signaling must be performed between involved nodes to inform them about successes and failures. A method for computing $T$ that considers all these contributions was proposed by Bratzik et al. [24]. For Deutsch's purification protocol [25] (which is characterized by $M \simeq 2^{(k+1)n}$), it results that $T \gg 2L_{tot}/c$, especially if probabilistic gates

are included. Therefore, $R$ easily falls below 1 Hz, for long distances (the upper bounds are illustrated in Figure 2, for different $k$, $n$, and $L_{tot}$ values).

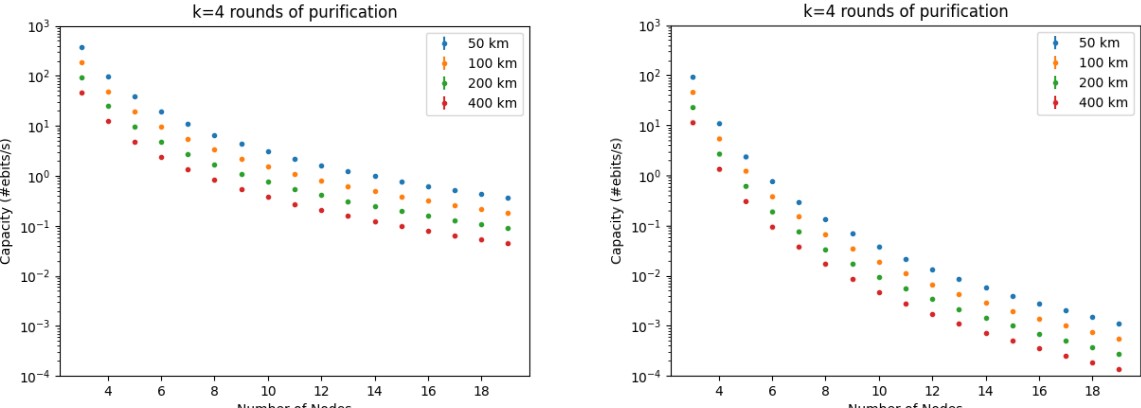

**Figure 2.** Capacity upper bounds for QR chains with entanglement swapping and $k$ rounds of purification, using Deutsch's protocol.

This poor performance is a consequence of the long-range two-way classical communication for purification and swapping operations, which may encompass the entire network diameter. Even if the swapping operations are made deterministic, the traditional purification protocols will remain probabilistic in nature [12].

To overcome this issue, purification protocols based on quantum error correction (QEC) have emerged in recent years. In these protocols, the quantum state of interest is encoded in multiple photons and the error correction performed at the repeater nodes can erase errors caused by photon loss and decoherence during transmission [11,26]. With QEC, entangled links can be purified while only requiring one-way classical communication [27], reason why $T = 2L_{tot}/c$.

Using a butterfly repeater design [27], the time to produce the long-range entangled pair scales as $T = O(2L/c)$, i.e., it is independent from the overall range $L_{tot}$. The memory resources depend on the QEC scheme, but in principle they scale as $M = O(\text{polylog}(L_{tot}))$ [12]. However, the near-deterministic generation of many-photon cluster states is required for encoding qubits, which is far beyond the state of the art [11,28]. Furthermore, there are stringent requirements on the control and readout fidelities within the repeater nodes. Theory research in this direction is promising [29], and experimental progress may bring such schemes closer to reality in the future.

According to Dhara et al. [30], using a QR chain with simultaneous BSMs, $M$ parallel channels (i.e., spatial or spectral multiplexing), and time-multiplexing block length $m$, the probability that the $i$-th BSM succeeds in at least one of the $Mm$ attempts is

$$P = 1 - (1 - \mu e^{-\alpha L_{tot}/N})^{Mm}, \tag{11}$$

where $N$ is the number of links (with length $L_{tot}/N$ each), $\mu$ is the linear-optical BSM efficiency, and $\alpha$ is the fiber loss coefficient. Therefore, the capacity of the QR chain is

$$R = \frac{q^{N-1}(1 - (1 - \mu e^{-\alpha L_{tot}/N})^{Mm})^N}{m\tau} \tag{12}$$

where $q$ is the probability of successful swap, and $\tau$ is the source repetition time. Some numerical results are illustrated in Figure 3. It can be proved that $R(L)$ traces a sub-exponential envelope when $m$ increases.

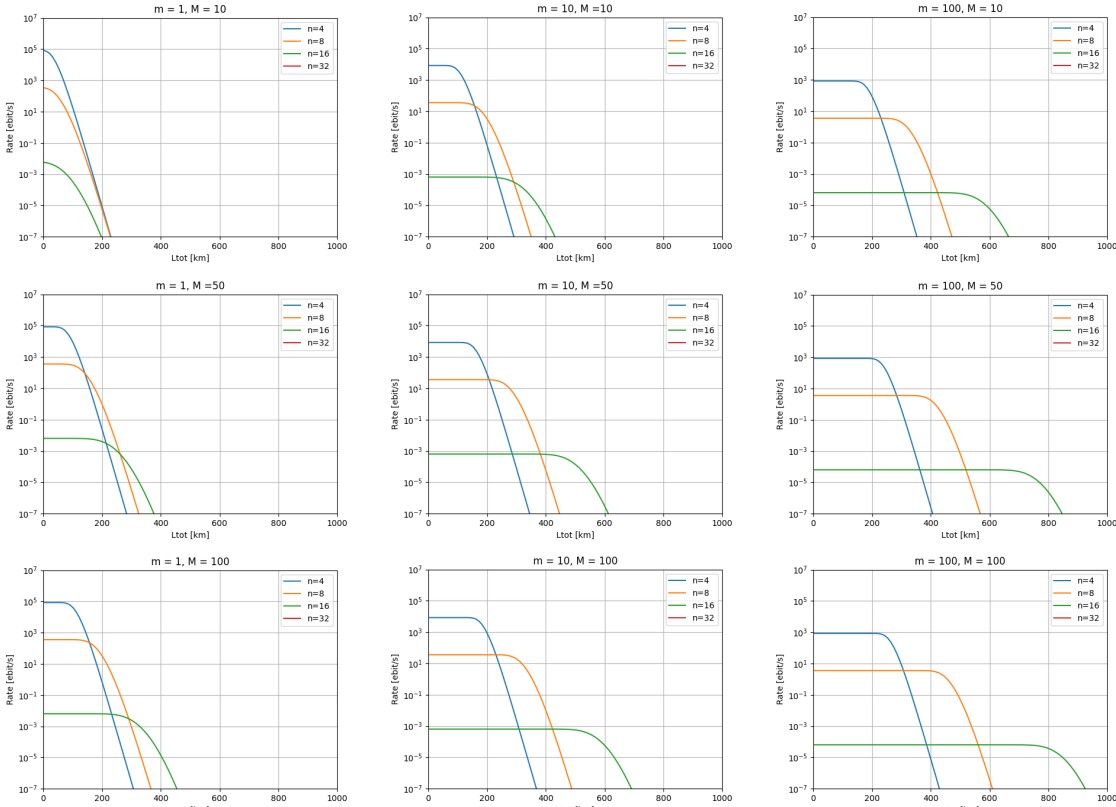

**Figure 3.** Capacity of a QR chain with simultaneous BSMs, using $M$ parallel channels, and a time-multiplexing block length $m$. Here, we assume $\alpha = 0.15$ dB/km, $\tau = 50$ ns, $\mu = 0.405$, and $q = 0.255$.

The aforementioned model can be adapted to the case of $N$ given optical fiber links with different lengths $L_i$ and loss coefficients $\alpha_i$, as formulated by the following theorem.

**Theorem 1.** *Let us consider a QR chain with simultaneous BSMs, $M$ parallel channels, time-multiplexing block length $m$, and $N$ given optical fiber links with lengths $L_i$ and loss coefficients $\alpha_i$ (where $i \in \{1, \ldots, N\}$). The capacity of the considered chain is*

$$R = \frac{q^{N-1}\Pi_{i=1}^{N}(1 - (1 - \mu e^{-\alpha_i L_i})^{Mm})}{m\tau}. \tag{13}$$

**Proof.** Generalizing Equation (11), the probability that the $i$-th BSM succeeds in at least one of the $Mm$ attempts is

$$P_i = 1 - (1 - \mu e^{-\alpha_i L_i})^{Mm}. \tag{14}$$

At the end of each $m\tau$ second block, when every QR node performs a BSM simultaneously, the probability that each of the $N$ links had heralded at least one Bell pair across it is

$$P_\pi = \Pi_{i=1}^{N}P_i. \tag{15}$$

Moreover, the probability that all the $N - 1$ BSMs at the QR nodes succeeded is $q^{N-1}$. Therefore, the capacity of the chain is

$$R = \frac{q^{N-1}P_\pi}{m\tau}. \tag{16}$$

This concludes the proof. $\square$

Some numerical results, concerning a 3-link QR chain, are illustrated in Figure 4. We observe that the capacity converges to a finite value, when $M$ and $m$ increase.

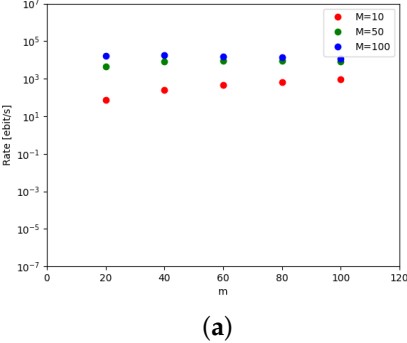 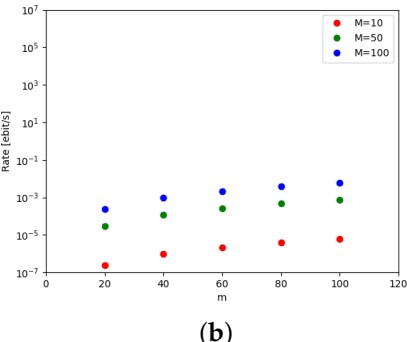

(**a**) (**b**)

**Figure 4.** Capacity of a QR chain with simultaneous BSMs, using $M$ parallel channels, and a time-multiplexing block length $m$. We assume a 3-link QR chain with $\alpha_1 = 0.25$ dB/km, $\alpha_2 = 0.15$ dB/km, $\alpha_3 = 0.25$ dB/km, $\tau = 50$ ns, $\mu = 0.405$, and $q = 0.255$. In (**a**), the three links have lengths $L_1 = 25$ km, $L_2 = 40$ km, and $L_3 = 30$ km, respectively. In (**b**), the lengths of all the three links are doubled.

### 3.4. Secret Key Rate for QKD

The core idea in QKD is to use quantum mechanics to detect the presence or absence of an eavesdropper, while Alice and Bob exchange quantum states with the purpose to create a shared secret key. In other words, Alice and Bob communicate over a (completely insecure) quantum channel. They also need a public classical authenticated channel in order to exchange data that are necessary to the correct execution of the protocol. The first and most famous QKD protocol, known as BB84, was proposed by Bennett and Brassard [31].

The *secret key rate* for a QKD protocol can be defined as

$$K = R r_\infty \tag{17}$$

where $r_\infty$ (secret fraction) is the ratio of secret bits and the measured bits in the asymptotic limit [24]. In the BB84 protocol, the upper bound of the secret fraction is

$$r_\infty^{BB84} = 1 - h(e_Z) - h(e_Y) \tag{18}$$

with $h$ being the binary entropy, and $e_Z$ and $e_Y$ being the error rates in the Z and Y bases, respectively [24].

## 4. Simulation-Based Performance Evaluation

It is not clear which QR technology will succeed, and there is no unique formular for modeling the end-to-end capacity due to different approaches. In this context, discrete event simulation is a robust tool that can strongly support QR network designers. In the following, we present a simple study of a QR chain based on the NetSquid simulation tool [32] as an illustrative example.

### 4.1. NetSquid

NetSquid [32] is one of the most advanced platforms for simulating quantum networks and modular quantum computing systems subject to physical non-idealities, ranging from the physical layer and its control plane up to the application layer. This is achieved by integrating several key technologies: a discrete-event simulation engine, a specialized quantum computing library, a modular framework for modeling quantum hardware devices, and an asynchronous programming framework for describing quantum protocols.

In NetSquid, five different formalisms for representing the quantum state are implemented, including the density matrix and the ket column vector ones, which are capable of universal quantum computing. Density matrices are more resource-consuming than ket vectors, but allow for representing mixed qubit states to simulate statistical ensembles or situations where the exact state is not known.

### 4.2. Simulation Model of a Quantum Repeater Chain

To simulate a QR chain, we consider equally spaced nodes, each holding a single quantum processor with 2 memory qubits. We refer to the outer nodes as end nodes. The in-between nodes are QRs. Two adjacent nodes are connected by a quantum link (modeled by an EntanglingConnection component) and by a classical link (modeled by a ClassicalConnection component). The sources of entanglement for adjacent nodes are characterized by a fixed timing delay (FixedDelayModel), while the optical links are characterized by a transmission delay model based on constant speed of photons through fiber (FibreDelayModel) and by a depolarization process (FibreDepolarizeModel) with depolarization probability

$$p = 1 - p_{di}10^{-L^2 p_{dl}/10} \tag{19}$$

where $p_{di}$ is the probability of depolarization on entering a fiber, $p_{dl}$ is the probability of depolarization per km of fiber, and $L$ is the distance between adjacent nodes.

The quantum processor is modeled by a QuantumProcessor component, whose qubits are affected by noise (T1T2NoiseModel, with relaxation time $T_1$ and dephasing time $T_2$). In detail [32], if a qubit in state $\rho$ is acted upon after having been idle for time $\Delta t$, a quantum process is applied such that

$$\rho \to E_0 \rho E_0^\dagger + E_1 \rho E_1^\dagger \tag{20}$$

where $E_0 = |0\rangle\langle 0| + \sqrt{(1-p)}|1\rangle\langle 1|$, $E_1 = \sqrt{p}|0\rangle\langle 1|$, and $p = 1 - e^{-\Delta t/T_1}$. Then, another quantum process is applied such that

$$\rho \to (1-p)\rho + pZ\rho Z, \tag{21}$$

where $Z = |0\rangle\langle 0| - |1\rangle\langle 1|$ and $p = (1 - e^{-\Delta t/T_2}e^{\Delta t/(2T_1)})/2$.

### 4.3. Results

The simulated QR chain performs entanglement swapping without purification. We used the density matrix representation and we performed 20 iterations for each configuration, to achieve statistical significance. In Figures 5 and 6, simulation results are reported, showing the fidelity of the end-to-end entangled pairs versus the number of nodes and the total distance.

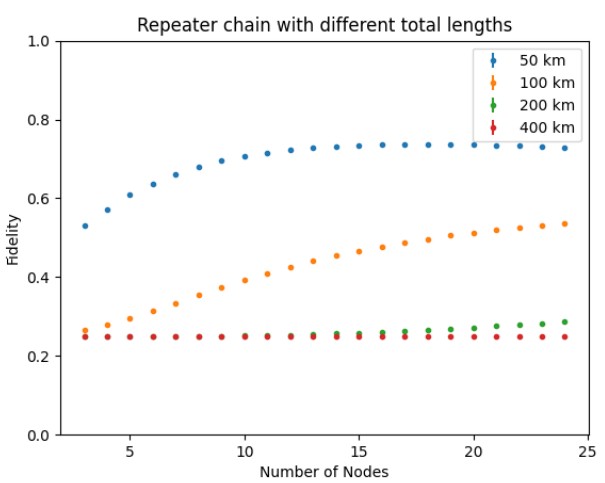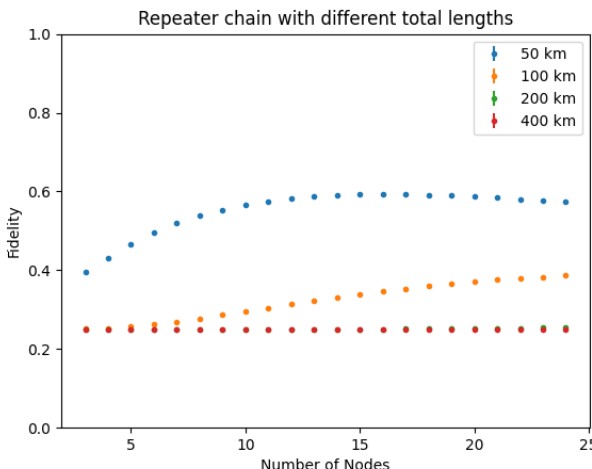

**Figure 5.** Fidelity vs. number of nodes, for different total lengths; (**left**) $p_{di} = 0.006$, $p_{dl} = 0.015$; (**right**) $p_{di} = 0.012$, $p_{dl} = 0.025$. In both cases, $T_1 = \infty$ and $T_2 = 1.46$ s. No purification is applied.

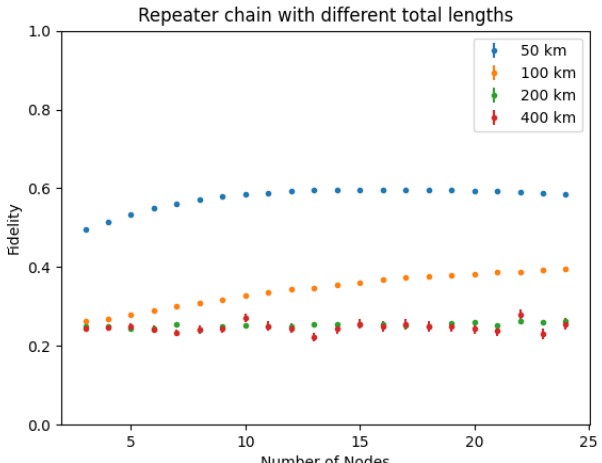 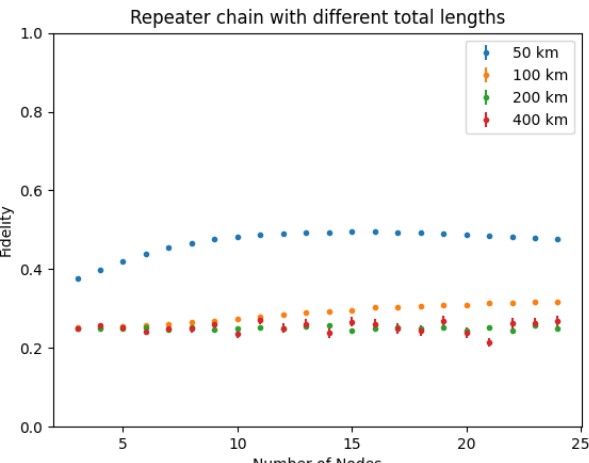

**Figure 6.** Fidelity vs. number of nodes, for different total lengths; (**left**) $p_{di} = 0.006$, $p_{dl} = 0.015$; (**right**) $p_{di} = 0.012$, $p_{dl} = 0.025$. In both cases, $T_1 = 2.68$ ms and $T_2 = 1$ ms. No purification is applied.

The effect of different values for the parameters of the depolarization process in the fiber links is illustrated. For the main parameters, the following values are considered:

- $c = 2 \times 10^8$ m/s
- $p_{di} = \{0.006, 0.012\}$
- $p_{dl} = \{0.015, 0.025\}$
- $T_1 = \{\infty, 2.68 \text{ ms}\}$
- $T_2 = \{1.46 \text{ s}, 1 \text{ ms}\}$

The adopted $T_1$, $T_2$ values are those used for the electron spin in nitrogen-vacancy quantum systems, as reported in [32].

## 5. Conclusions

Today, a second quantum revolution is underway, leveraging on the quantum principles of superposition, entanglement and measurement. Four main application areas of quantum technologies and services have been identified so far: quantum secure communications, quantum computing, quantum simulations, quantum sensing and metrology.

Assumed that management and control of quantum nodes is a key challenge under definition, today, a main obstacle in exploiting long-range QKD, Quantum-ready networks, and then (in the long term) Quantum Internet, concerns the inherent losses due to the optical transmission channels. Today, it is assumed that the most promising way for overcoming this limitation, while avoiding costly trusted node configurations, is based upon the adoption of Quantum Repeaters.

In this paper, we focused on quantum communications based on entanglement distribution. In particular, we offered a brief introduction on the evolution towards the Quantum Internet, with particular reference to the issues under study, such as quantum transmission, networking protocols and management-control paradigms. Then, we provided an overview of current strategies for end-to-end entanglement generation, with the discussion of capacity upper bounds. We adapted the model proposed by Dhara et al. [30] regarding a QR chain with simultaneous BSMs, parallel channels and time-multiplexing, to the case of $N$ given optical fiber links with different lengths. We observed that the capacity converges to a finite value, when the number of channels and the time-multiplexing block length increase. Moreover, we discussed the impact of capacity upper bounds and link lengths between QRs on the secret key rate in QKD systems. Finally, we presented the simulation of a simple quantum network model with a certain number of QRs using the NetSquid simulation tool.

The considered models and results are still preliminary and further investigations are required: nevertheless, it is argued that simulations are very important for analyzing the

performance of quantum networks and, as such, for optimizing the architectural design towards future networks and the Quantum Internet. Importantly in this architectural definition the protocols evolution, the management and control requirements should be kept under consideration "by design".

Indeed, analytical models become very complex when all the design parameters for end-to-end entanglement generation and realistic error models are to be considered. Advanced tools like NetSquid are highly suitable for the simultaneous evaluation of metrics like quantum state fidelity, QR capacity, and secret key rate, which are fundamental indices to design quantum secure communications services.

**Author Contributions:** Conceptualization, A.M. and M.A.; methodology, A.M. and M.A.; software, M.A.; validation, M.A.; formal analysis, M.A.; investigation, A.M. and M.A.; data curation, M.A.; writing—original draft preparation, A.M. and M.A.; writing—review and editing, A.M. and M.A.; supervision, A.M.; project administration, A.M.; funding acquisition, A.M. All authors have read and agreed to the published version of the manuscript.

**Funding:** This research received no external funding.

**Institutional Review Board Statement:** Not applicable.

**Informed Consent Statement:** Not applicable.

**Conflicts of Interest:** The authors declare no conflict of interest.

## Abbreviations

The following abbreviations are used in this manuscript:

BSM  Bell-State Measurement
QKD  Quantum Key Distribution
QR  Quantum Repeater

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
