# Peer review of "End-to-End Entanglement Generation Strategies: Capacity Bounds and Impact on Quantum Key Distribution"

_quantumrep, doi:10.3390/quantum4030017_

Round 1

Reviewer 1 Report

In the manuscript quantumrep-1816600, entitled “End-to-End Entanglement Generation Strategies: Capacity Bounds and Impact on Quantum Key Distribution”, the authors provide a brief introduction on the evolution towards the Quantum Internet and an overview of current strategies for end-to-end entanglement generation with the discussion of capacity upper bounds and their impact of secret key rate in QKD. They also presenting the simulation of a simple quantum network model with a certain number of quantum repeaters using the NetSquid simulation tool. Recently, the Quantum Internet has received a lot of attention. The authors' work can sort out the understanding of the aspect on entanglement generation strategies. I can recommend its publication provided the authors can consider the following comments:

 1. Research papers on the quantum key distribution should be added in the Introduction, such as references [Entropy 24, 460 (2022); Journal of the Optical Society of America B 36, B83-B91 (2019); Physics Letters A 381, 1393-1397 (2017); Physical Review A 94, 032335 (2016); Physical Review A 88, 052322 (2013)].

  2. In Figs 5 and 6, the figure legends are inconsistent with the representation of data lines.

Author Response

  1. Research papers on the quantum key distribution should be added in the Introduction, such as references [Entropy 24, 460 (2022); Journal of the Optical Society of America B 36, B83-B91 (2019);Physics Letters A 381, 1393-1397 (2017); Physical Review A 94, 032335 (2016); Physical Review A 88, 052322 (2013)].

Response:

In the Introduction, we have added several references, including those to the suggested research papers [1-5].

  1. In Figs 5 and 6, the figure legends are inconsistent with the representation of data lines.

Response:

The captions of figures 5 and 6 have been modified to make them consistent with the represented data lines. In all the plots, the legends indicate the different values of the considered total length (50, 100, 200 and 400 km).

Reviewer 2 Report

End-to-End Entanglement Generation Strategies: Capacity Bounds and Impact on Quantum Key Distribution

The subject of the paper is End-to-End Entanglement Generation Strategies: Capacity Bounds and Impact on Quantum Key Distribution. This paper focuses on quantum secure communications by addressing the evolution of Quantum Key Distribution (QKD) networks. The network is based on the entanglement generation between nodes. However, some recommendations should be taken into account for publication:

Originality/Novelty

It shown some result from this paper that acceptable for publication.

Quality of Presentation

The overall presentation quality is acceptable. The article is, on the whole, well-written; the English used in the paper is understandable. In line 119 some repeated word need to be remove.

-          Methodology

The method presented is not clear as can be seen in Figure 1. It needs more clear explanation of generation of entaglement state between the nodes.

-          Results and Discussion

The result show an intresting finding about the fidelity of quantum entanglement state with nodes used. It also shown the distance between the nodes and its effect towards the rate of ebits.

Conclusions

The conclusions are straightforward. This needs to be improved. Include more specifics about the outcomes that were achieved.

Overall Merit:

The article certainly has some merit. For the rest, I believe that the article is organised in a logical and understandable manner.

Author Response

In line 119 some repeated word need to be remove.

Response:

In line 119, the repetition of the word “error” has been fixed.

The method presented is not clear as can be seen in Figure 1. It needs more clear explanation of generation of entaglement state between the nodes.

Response:

Figure 1 and its caption have been updated, in order to better illustrate the generation of entanglement beetween the nodes.

The conclusions are straightforward. This needs to be improved. Include more specifics about the outcomes that were achieved.

Response:

We revised the conclusions, providing a more precise summary of the outcomes that we achieved, regarding the capacity of Quantum Repeter chains and its impact on the secret key rate in QKD systems.